# Steroid-Refractory Immune-Related Adverse Events Induced by Checkpoint Inhibitors

**DOI:** 10.3390/cancers15092538

**Published:** 2023-04-28

**Authors:** Dirk Tomsitz, Theresa Ruf, Sarah Zierold, Lars E. French, Lucie Heinzerling

**Affiliations:** 1Department of Dermatology and Allergy, University Hospital, LMU Munich, 80539 Munich, Germany; 2SERIO Side Effects Registry Immunooncology, 80337 Munich, Germany; 3Department of Dermatology and Cutaneous Surgery, University of Miami Miller School of Medicine, Miami, FL 33136, USA; 4Department of Dermatology and Allergy, University Hospital Erlangen, 91054 Erlangen, Germany

**Keywords:** steroid-refractory immune-related adverse events, steroid-dependent immune-related adverse events, skin cancer, second-line immunosuppression

## Abstract

**Simple Summary:**

With the common use of immune checkpoint inhibitors (ICIs) as a standard therapy for many tumor entities, the number of immune-related adverse events (irAEs) has increased. Side effect management for irAEs includes the administration of corticosteroids which are mostly very effective. However, a subgroup of patients with checkpoint inhibitor-induced side effects do not adequately respond to steroids with so-called steroid-refractory side effects (sr-irAEs) or cannot be tapered off steroids without the recurrence of side effects, the so-called steroid-dependent side effects (sd-irAEs). Since little is known about the incidence and management of sr/sd-irAEs, we investigated the occurrence and management of these difficult-to-treat side effects. This is the first study to report on the incidence rate of steroid-refractory or steroid-dependent irAEs in patients with skin cancer.

**Abstract:**

The occurrence, second-line management and outcome of sr/sd-irAEs was investigated in patients with skin cancer. All skin cancer patients treated with immune checkpoint inhibitors (ICIs) between 2013 and 2021 at a tertiary care center were analyzed retrospectively. Adverse events were coded by CTCAE version 5.0. The course and frequency of irAEs were summarized using descriptive statistics. A total of 406 patients were included in the study. In 44.6% (n = 181) of patients, 229 irAEs were documented. Out of those, 146 irAEs (63.8%) were treated with systemic steroids. Sr-irAEs and sd-irAEs (n = 25) were detected in 10.9% of all irAEs, and in 6.2% of ICI-treated patients. In this cohort, infliximab (48%) and mycophenolate mofetil (28%) were most often administered as second-line immunosuppressants. The type of irAE was the most important factor associated with the choice of second-line immunosuppression. The Sd/sr-irAEs resolved in 60% of cases, had permanent sequelae in 28% of cases, and required third-line therapy in 12%. None of the irAEs were fatal. Although these side effects manifest in only 6.2% of patients under ICI therapy, they impose difficult therapy decisions, especially since there are few data to determine the optimal second-line immunosuppression.

## 1. Introduction

Immune checkpoint inhibitors (ICIs) have become standard therapy for many tumor entities. They target the PD1-/PD-L1 pathway, the CTLA-4 pathway, or the LAG-3 pathway and are either used as monotherapy or in combination, e.g., with different ICI or chemotherapy [1]. Along with the improved clinical outcome, severe toxicities or so-called immune-related adverse events (irAEs) are induced. Grade 3 and 4 side effects were observed in 6–42% of patients treated with anti-PD1 or anti-PDL1-antibodies [2,3,4,5], in 28% of patients treated with ipilimumab [6], in 59% of patient treated with combined ipilimumab and nivolumab [6], and in 19% of patients with combined relatlimab and nivolumab according to CTCAE version 5.0 [7].

Systemic corticosteroids are considered the first-line therapy for the management of irAEs in most organ systems [8,9,10]. However, a subgroup of patients with irAEs does not adequately respond to steroids and have so-called steroid-refractory side effects (sr-irAEs), or cannot be tapered off steroids without the recurrence of side effects, called steroid-dependent side effects (sd-irAEs). In sr-irAEs and sd-irAEs, various second-line immunosuppressants have been suggested and used. Depending on the affected organ, recommendations range from classical systemic immunosuppressants such as mycophenolate mofetil, cyclosporine A, methotrexate or azathioprine, monoclonal antibodies targeting tumor necrosis factor alpha (TNF-α, infliximab), α4β7-integrin (vedolizumab), or CD20 (rituximab), and intravenous immunoglobulins, antithymocyte globulin or plasmapheresis. There are little data available on the immunologic mechanisms of irAEs to date, however, CRP and interleukin 6 (IL-6) are clearly upregulated [11], and the anti-IL6 receptor antibody tocilizumab was also shown to be effective [12].

Alemtuzumab, an anti-CD52 monoclonal antibody, was recently shown to be effective in a single case of immune-related myocarditis [13], and abatacept, a CTLA-4 agonist which leads to an inhibition of CD28-B7-mediated T-cell costimulation, has also shown potential [14]. In a patient with immune-related colitis refractory to corticosteroids, infliximab and cyclosporine in combination with extracorporeal photopheresis led to the resolution of symptoms and the expansion of natural killer cells with an immunoregulatory phenotype [15]. Two patients with irColitis, who failed to respond to both systemic steroids and vedolizumab, were effectively treated with the anti-IL12/IL23 antibody ustekinumab [16]. The blockade of IL17A with secukinumab successfully led to the improvement in pain and swelling in two patients with immune-related inflammatory arthropathy and to resolution in one patient with a psoriasiform exanthema which worsened after tapering systemic steroids [17,18]. However, to date, management decisions are mostly based on expert opinion [19], since prospective or comparative data on the outcome of second-line immunosuppression are lacking. Even current treatment recommendations for irAEs contained in the NCCN, ASCO, and ESMO guidelines are based on clinical experience.

Clearly, immunosuppression hampers the tumor response. While Schadendorf at al. showed no difference in progression-free survival between patients who discontinued treatment with ICI due to irAEs and patients who continued ICI-treatment in a recent study [20], patients treated with high-dose corticosteroids (≥60 mg/day) for their irAE(s) showed worse progression-free survival and overall survival compared to patients not treated with high-dose corticosteroids [21]. Additionally, in patients who were managed with TNF-inhibitors for sr-irAEs, survival was significantly decreased compared to the patients treated with corticosteroids only [22]. In a murine study, concomitant treatment with a TNF-α blocker and combined anti-PD-1 and anti-CTLA4 antibodies ameliorated colitis and improved the anti-tumor efficacy [23]. This led to a prospective clinical study investigating the use of the preventive therapy of infliximab or certolizumab pegol in combination with combined immunotherapy with ipilimumab and nivolumab [24]. Other prospective studies investigating the efficacy of tofacitinib, a Janus kinase inhibitor, in patients with sr-irColitis (NCT04768504), or rituximab and tocilizumab in patients with sd-irAEs (NCT04375228). New substances, such as CD24Fc, which binds to injured cell components and prevents inflammatory responses, are also under investigation for the management of irAEs (NCT04552704).

Recently, in a retrospective monocentric study including 2750 patients with lung cancer who were treated with ICI, 51 (1.9%) experienced sr-irAEs and were managed with an additional immunosuppressant [25]. The majority were treated with TNF-inhibitors (73%) or mycophenolate mofetil (20%). Unfortunately, data on the frequency, management, and outcome of sr/sd-irAEs have not been analyzed to date within the prospective clinical trials, or at least not published. No data exist for sr/sd-irAEs in skin cancer patients.

In this study, the occurrence of sr/sd-irAE side effects in patients with skin cancer was investigated in a real-world setting. Additionally, second-line management was compared with treatment algorithms recommended in the management guidelines.

## 2. Materials and Methods

Between January 2013 and December 2021, all patients who were treated with an ICI (avelumab, cemiplimab, ipilimumab, nivolumab, pembrolizumab, and/or combined ipilimumab and nivolumab) at a tertiary care university skin cancer center in Munich, Germany, were retrieved from Side Effect Registry Immuno-Oncology (SERIO), an international registry based at the Ludwig-Maximilian University Hospital in Munich that collects cases of irAEs in cooperation with the Paul Ehrlich Institute. The registry contains anonymous data on all patients who were treated with ICI from our department, therefore, informed consent was not mandatory. Once they received systemic tumor therapy, the patients were followed-up at our center until death or long term.

Patients were treated for advanced/metastatic non-melanoma skin cancer and melanoma (including melanoma of unknown primary, uveal melanoma and mucosal melanoma) in an adjuvant setting or as a neoadjuvant therapy. The individual electronic patient files of these patients were screened for the occurrence of irAE, type of irAE, severity, and outcome. At our center, patients are surveyed for potential adverse events (including fatigue for hypophysitis, diarrhea, dyspnea for pneumonitis), physically assessed at each visit (neurological side effect), and their laboratory values are checked (e.g., transaminases for hepatitis) before each infusion. Additionally, the patients received an emergency card with a telephone contact available 24/7 to call in case of side effects. Side effects were attributed to the medication by physicians and grading performed according to the Common Terminology Criteria for Adverse Events (CTCAE) version 5.0.

IrAE data (type of irAE, severity of irAE, onset of irAE after initiation of ICI therapy) the first-line and second-line management of irAE, time to first response, and outcome of irAE were collected (Figure 1). Second-line treatments were compared with management recommendations for the treatment of irAEs from ASCO, NCCN, and ESMO guidelines. Descriptive statistics were used to summarize the frequencies and courses of irAEs.

At our center, the decision of systemic tumor therapy is based on the BRAF/c-kit mutation status (for targeted therapy), performance status, and co-morbidities. If there are several treatment alternatives, patients present the pros and cons in terms of efficacy and toxicity to determine the patient preferences. Furthermore, potential study options (e.g., cellular therapies) are presented to the patient.

Continuous data are presented as a median or ranges and categorical data are presented as percentages. Overall survival (OS) was compared between patients with regard to the development, severity and management of irAEs. Log-rank tests were performed to compare OS between two groups (patients with irAEs versus patients without irAE, patients with grade 1/2 irAEs versus patients with grade 3/4 irAEs, and patients with grade 3/4 irAEs who were treated with corticosteroids only versus patients who were treated with second-line immunosuppressants). *p* values < 0.05 were considered statistically significant. Statistical analyses were conducted with SPSS Version 27.

The institutional review board of the medical faculty of the Munich University Hospital (dating 17 February 2021; 20-1122) approved the data analysis from the SERIO registry. This study was conducted in accordance with the principles of the Helsinki Declaration.

## 3. Results

### 3.1. Patients’ Characteristics

In total, 406 patients (234 males and 172 females) who were treated with ICIs were identified from our registry. The majority of patients were treated for melanoma (n = 389, 95.8%; including cutaneous melanoma, mucosal melanoma, conjunctival melanoma, and uveal melanoma). Other patients had non-melanoma skin cancer (squamous cell carcinoma, basal cell carcinoma, Merkel cell carcinoma, and eccrine porocarcinoma). In 76.4% of cases, the ICI treatment was performed in patients with advanced/metastatic disease, while 21.2% received ICI as an adjuvant therapy after the complete resection of the tumor or were treated in a neoadjuvant setting (0.2%). Most patients (n = 217, 53.4%) were treated with an anti-PD1/PDL1 antibody (cemiplimab, nivolumab, pembrolizumab, or avelumab) or combined anti-CTLA4-antibody and anti-PD1 antibody (n = 173, 42.6%; ipilimumab and nivolumab). In a total of 181 (44.6%) patients, at least one irAE was documented in the patient chart (Figure 1, Table 1).

### 3.2. Distribution and Classification of irAEs

Among the 406 patients treated with ICIs, 181 patients (44.6%) developed at least one irAE and 8.9% developed more than one (two irAEs n= 29, three irAEs n = 6, four irAEs n = 1), but altogether 229 individual irAEs were documented. Regarding organ involvement, 50.0% (n = 114) of the toxicities were located in the gastrointestinal tract with colitis (n = 50, 21.8%) and hepatitis (n = 44, 19.2%) being the most frequent, followed by pancreatitis (n = 16, 7.0%), gastritis (n = 3, 1.3%), and cholecystitis (n = 1, 0.4%). Endocrine toxicities represented the second largest group with 18.3% (n = 42), and comprised thyroiditis (n = 18, 7.8%), hypophysitis (n = 19, 8.3%), adrenalitis (n = 4, 1.7%), and type 1 diabetes mellitus (n = 1, 0.4%). Cutaneous irAEs were the third largest group (n = 22, 9.6%), and included maculopapular exanthema (n = 16, 7.0%), lichenoid skin eruption (n = 4, 1.7%), bullous skin changes (n = 2, 0.9%), and alopecia (n = 1, 0.4%). Pulmonary (n = 12, 5.2%), neurologic (n = 11, 4.8%), rheumatoid (n = 10, 4.4%), cardiac (n = 7, 3.1%), renal (n = 6, 2.6%), and ophthalmic irAEs (n = 3, 1.3%) were less frequently detected (Figure 2).

The severity of the irAEs was classified as CTCAE grade 1 in 12.2% (n = 28), grade 2 in 43.2% (n = 99), grade 3 in 31.0% (n = 71), and grade 4 in 9.6% (n = 22) of cases, with no grade 5. In 3.9% (n = 9) of cases, we were not able to classify according to CTCAE version 5.0.

The onset of neurologic toxicities, nephritis, colitis, and thyroiditis were documented early after beginning ICI treatment with a mean of 2.1 months (range 1.5–5 months), 2.7 months (range 0.25–4 months), 2.8 months (range 0.07–26 months), and 2.8 months (range 0.25–11 months), respectively. On the other hand, pneumonitis, pancreatitis, and ophthalmic toxicities occurred late after the initiation of ICI therapy with a mean onset after 6.4 months (range 1–24 months), 6.6 months (range 0.5–25 months), and 9.6 months (range 9–11 months), respectively. Other irAEs were documented between 3–6 months after first ICI dosage (Table 2).

### 3.3. First-Line Management and Outcome of irAEs

First-line treatment of irAEs was performed using systemic corticosteroids in 63.8% (n = 146) of cases and ICI-treatment was continued despite the onset of irAEs without any specific treatment in 7.8% (n = 18). For endocrine toxicities, systemic corticosteroids were usually not administered, but the defective hormone production was addressed by the substitution of the corresponding hormone (n = 38, 16.6%). For the treatment of irAEs affecting easily accessible organs, local treatment was performed. This was the case for topical corticosteroids used for skin toxicities and conjunctivitis (n = 18, 7.9%), intravitreal corticosteroids used for ophthalmic toxicities (n = 1, 0.4%), inhalative corticosteroids used for pulmonary irAEs (n = 1, 0.4%), and intralesional/intraarticular corticosteroids combined with systemic corticosteroids used for neurologic toxicities or arthritis (n = 3, 1.3%). In eight cases (3.5%), ICI therapy was stopped due to the occurrence of an irAE without the initiation of a specific treatment.

Complete resolution after first-line treatment was documented in 57.2% (n = 131), whereas no resolution but the improvement of symptoms was seen in 13.1% (n = 30), and no improvement at all occurred in 2.6% (n = 6). Permanent hormone substitution after the insufficiency of endocrine glands was required in 16.2% (n = 37). Second-line immunosuppressants were required in 10.9% (n = 25) of all side effects.

In responsive adverse events, a significant improvement was first detected between a period of one and 56 days. In irAEs, which were treated by hormone substitution, time to first improvement was shortest with a mean of 1.0 day for irDiabetes, 1.0 day for irAdrenalitis, 1.9 days for irHypophysitis, and 2.4 days for irThyroiditis. The longest treatment response was observed in ophthalmic toxicities that were treated with local corticosteroids with a mean of 8.0 days, and in cutaneous toxicities which were treated with topical corticosteroids with a mean of 9.5 days (Table 2).

### 3.4. Incidence of sr-/sd-irAEs

Among the patients that were treated with systemic steroids as a first-line therapy for irAEs, the highest rate of sr-/sd-irAEs of 77.8% (n = 7) was documented in patients with irArthritis, followed by 75.0% in patients with dermatologic toxicities (n = 3), which comprised a lichenoid rash and bullous drug reaction, whereas maculopapular exanthema was well managed with topical steroids. The most frequent irAEs which were treated with systemic steroids in our cohort were steroid-refractory or -dependent irPneumonitis in 40.0% (n = 4), irColitis in 34.9% (n = 15), and irHepatitis in 26.8% (n = 11, Table 3).

### 3.5. Second-Line Management

In 25 patients (10.9%), second-line immunosuppression was initiated after the insufficient efficacy of steroid therapy. Infliximab was used most frequently (48% of patients, n = 12), and the indication was for irColitis. Patients with irHepatitis were treated with mycophenolate mofetil, representing 28% of second-line therapies (n = 7). Two patients (8%) with irArthritis were treated with methotrexate. In cutaneous toxicities, a lichenoid rash was treated with acitretin (n = 1) and extracorporeal photopheresis (n = 1), and bullous drug reactions or immune-related bullous pemphigoid (irBP, n = 1) was treated with intravenous immunoglobulins. One patient with irMyocarditis was treated with intravenous immunoglobulins in addition to high-dose corticosteroids.

The time to response was shortest in patients treated with infliximab for irColitis with a mean time of 1.3 days, and in patients with irHepatitis, who were treated with mycophenolate mofetil with a mean time to improvement of 2.8 days. Other second line-therapies took longer to induce an improvement in symptoms, with latencies ranging from 2 weeks to 3 months (Table 4).

In three patients, a subsequent therapy after non-responsiveness to second-line management was initiated: (1) a patient with grade 4 irColitis, who did not respond to two doses of infliximab at all, was treated with 3 doses of the α_4_β_7_ integrin blocker vedolizumab and high doses glucocorticoids until a first reduction in stool frequencies and pain was achieved. (2) A second patient with grade 4 irColitis, who also did not respond to two doses of infliximab, was treated with a combination of extracorporeal photopheresis and two doses of vedolizumab. Here, a first significant improvement was documented after 6 weeks. (3) A patient who suffered from biopsy-proven grade 4 irHepatitis and who did not show any improvement after the initiation of mycophenolate mofetil 2 mg/kg was switched to the calcineurin inhibitor tacrolimus (10 ng/mL whole blood trough concentration) and eventually to sirolimus (10 ng/mL whole blood trough concentration) when the first significant reduction in liver enzymes occurred after 6 months. Even 5 years after the onset of irHepatitis, it was still not possible to completely taper sirolimus and prednisolone.

### 3.6. Survival

To investigate the influence of irAE occurrence and side effect management on survival, we compared the OSs of patients who developed irAEs (n = 125) with those of patients without irAEs (n = 117). For better comparability, only patients with metastatic cutaneous melanoma (n = 242) were taken into account.

Although not statistically significant, patients without irAEs had a longer mean OS of 90.0 months (95% CI 76.2 to 103.8) compared to 69.3 months (95% CI 59.5 to 79.1) in patients who developed irAEs (*p* = 0.780).

We then analyzed the impact of irAE severity on OS. Patients with mild/moderate irAEs (Grade 1 and 2 according to CTCAE v 5.0, n = 66) had a longer mean OS of 75.1 months (95% CI 62.3–88.0) compared to 47.4 months (95% CI 37.9–56.8) in patients with severe irAEs (Grade 3 and 4, n = 59, *p* = 0.135).

Regarding the management of severe irAEs (Grade 3 and 4), patients who were treated with a second-line immunosuppressant (n = 16) had a longer mean OS of 56.2 months (95% CI 40.5 to 71.9) compared to 45.6 months (95% CI 34.4–56.8) in patients who were only treated with corticosteroids (n = 43, n = 0.321, Figure 3).

### 3.7. Comparison with Guidelines

Second-line treatment of irColitis with infliximab, irHepatitis with mycophenolate mofetil, and irArthritis with methotrexate was in accordance with all three guidelines. Intravenous immunoglobulins for the therapy of myocarditis are covered along with antithymocyte globulin, infliximab, and mycophenolate mofetil by NCCN guidelines, whereas ASCO guidelines recommend therapy with mycophenolate mofetil, infliximab or antithymocyte globulin, and ESMO guidelines recommend tocilizumab, mycophenolate mofetil, antithymocyte globulin, alemtuzumab, or abatacept. Recommendations for steroid-refractory or steroid-dependent skin toxicities are given by ASCO and ESMO guidelines: rituximab in bullous dermatoses and intravenous immunoglobulins or cyclosporine for severe cutaneous dermatoses (ASCO guidelines), and infliximab or tocilizumab for a maculopapular rash (ESMO guidelines). Guidance on the lichenoid rash is not included. Acitretin and extracorporeal photopheresis for the treatment of lichenoid rash and intravenous immunoglobulins for the treatment of bullous dermatoses were not contained in any guideline.

For second-line management in our patient cohort, there was 84% (n = 21) accordance with ASCO guidelines, 88% (n = 22) accordance with NCCN guidelines, and 84% (n = 21) accordance with ESMO guidelines (Table 5).

## 4. Discussion

Our study shows a remarkable incidence of 6.2% of patients with steroid-refractory and steroid-dependent side effects under ICI therapy in a large cohort of skin cancer patients. In a total of 406 patients, 10.9% of all documented immune-related side effects did not sufficiently respond to steroids or became exacerbated when trying to taper the steroid therapy.

Compared to the data of Luo et al., who detected 51 patients (1.9%) with sd/sr irAEs among 2750 patients with lung cancer, the frequency in our patient cohort was more than three-fold higher [25]. Whereas the incidence of irColitis was comparable in both groups (48% vs. 53%), the frequency for hepatitis was lower in the cohort of Luo et al. (28% versus 12%), which could be due to the differences in monitoring transaminases or the lower percentage of patients treated with combined ipilimumab and nivolumab. Our cohort had 0% sd/sr-pneumonitis compared to 20% in the lung cancer group. The severity of pneumonitis is known to be higher in lung cancer patients compared to other cancer types, as risk factors such as smoking, previous lung disease, and prior radiotherapy or tyrosine kinase inhibitor therapy are more frequent in this population [26]. Importantly, up to 52% of checkpoint inhibitor-induced irAEs or their sequelae persist after the cessation of ICI therapy [27,28]. Thus, this special toxicity profile has to be taken into account when deciding between immunotherapy or targeted therapy in which side effects are reversible. Currently, the benefit of ICI outweighs toxicity. However, if severe toxicity could be predicted, this would be extremely valuable, especially when treating patients in early disease stages where the absolute risk reduction is smaller [29]. This approach is not yet successful, however. Thus, current efforts are aimed at continuing to reduce exposure to ICIs by combining them with the mRNA vaccine [30].

The higher rate of sr/sd-irAEs in this study can be explained by the high frequency of combined immunotherapy with ipilimumab and nivolumab (43% compared to 37%), known to induce >grade 3 toxicity in 59% [6]. Regarding the outcome of first-line therapies for irAEs, overall, 13.1% of irAE symptoms improved without the complete resolution of the irAEs, and in 2.6%, no improvement was induced. Retrospectively, we estimate that a second-line immunosuppressive treatment should have been started earlier in those patients and more frequently since prompt treatment has been shown to improve the response [31].

In our cohort, patients with metastatic cutaneous melanoma treated with corticosteroids plus second-line immunosuppressants showed a trend towards a longer OS compared to patients treated with corticosteroids only for grade 3/4 irAEs. This is in accordance with data from another retrospective multicenter cohort study which included 771 patients with metastatic melanoma [32].

Nowadays, second-line immunosuppression is indicated early in steroid-refractory irAEs. For instance, according to the NCCN guidelines, if no improvement is noted within 2–3 days of intravenous methylprednisolone at a dose of 1–2 mg/kg per day, the addition of infliximab or the α_4_β_7_ integrin blocker vedolizumab is strongly recommended. Similarly, additional treatment with mycophenolate mofetil is recommended for steroid-refractory hepatitis after 3 days of initial treatment [8]. Patients in our study responded within a mean time of 1.3 days after the initiation of infliximab for irColitis, and 2.8 days after the initiation of mycophenolate mofetil for irHepatitis. On the contrary, the time to first response ranged from 2 weeks to 3 months for other steroid-refractory irAEs.

The limitations of our study include its retrospective nature that potentially underestimates the frequency of irAEs. Furthermore, monocentric data may not be applicable to multicentric cohorts of patients.

Treatment recommendations have been established by the ASCO, NCCN, and ESMO. However, these guidelines are not based on studies comparing different treatment approaches. In our patients, accordance with the recommendations in the published guidelines ranged between 76% and 88%. Management data after the failure of first-line treatment for some irAEs, such as lichenoid rash and bullous drug reactions in our population, are lacking.

## 5. Conclusions

This study assessed the frequency of steroid-refractory or steroid-dependent irAEs in patients with cutaneous malignancies who were treated with ICIs. Prospective trials with larger patient populations or analyses of combined clinical study data are needed to gain a better understanding of rare steroid-refractory or steroid-dependent toxicities.

## Figures and Tables

**Figure 1 cancers-15-02538-f001:**
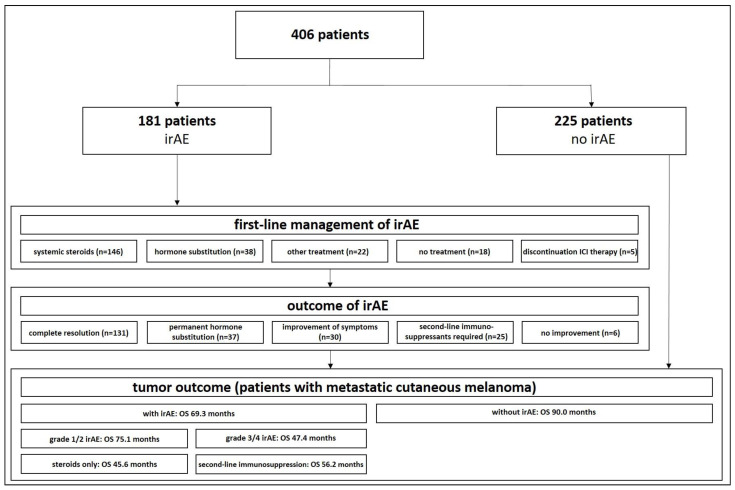
Study overview.

**Figure 2 cancers-15-02538-f002:**
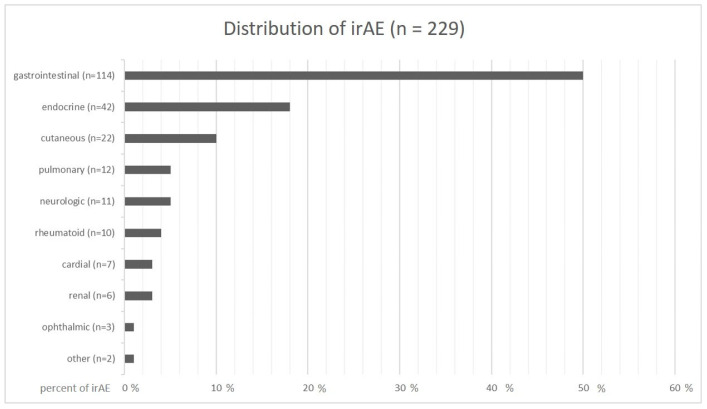
Distribution of irAEs.

**Figure 3 cancers-15-02538-f003:**
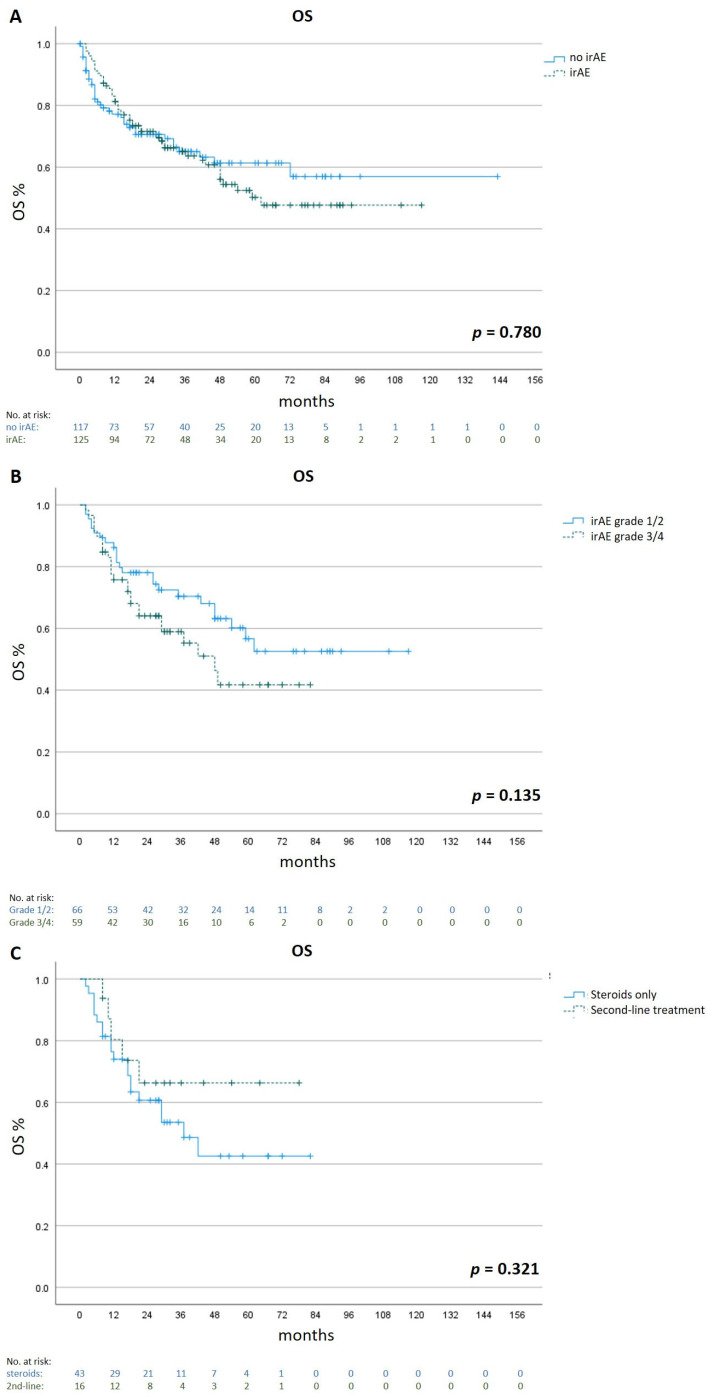
Overall survival in different subgroups. Kaplan–Meier estimates were calculated for the OSs of patients who developed irAEs (n = 125) and patients who did not develop irAEs (n = 117, Panel **A**), for the OSs of patients with mild/moderate irAEs (n = 66) and severe irAEs (n = 59, Panel **B**), and for the OSs of patients with severe irAEs who were treated with corticosteroids only (n = 43) or corticosteroids plus second-line immunosuppressants (n = 16, Panel **C**). Only patients with metastatic cutaneous melanoma were included.

**Table 1 cancers-15-02538-t001:** Characteristics of the patients.

	Patients
	n = 406
Age—years	
Mean	64.0
Range	22–91
Sex—no. (%)	
Male	234 (57.6)
Female	172 (42.4)
Tumor type—no. (%)	
Cutaneous melanoma	296 (72.9)
Melanoma of unknown primary	45 (11.1)
Mucosal melanoma	13 (3.2)
Conjunctival melanoma	2 (0.5)
Uveal melanoma	33 (8.1)
Cutaneous squamous cell carcinoma	11 (2.7)
Eccrine porocarcinoma	1 (0.2)
Basal cell carcinoma	2 (0.5)
Merkel cell carcinoma	3 (0.7)
Treatment setting—no. (%)	
Neoadjuvant	10 (0.2)
Adjuvant	86 (21.2)
Metastatic	310 (76.4)
Treatment—no. (%)	
Avelumab	1 (0.2)
Cemiplimab	13 (3.2)
Ipilimumab	15 (3.7)
Nivolumab	75 (18.5)
Pembrolizumab	129 (31.8)
Ipilimumab + nivolumab	173 (42.6)
Occurrence of irAE—no. (%)	
No	225 (55.4)
Yes	181 (44.6)

**Table 2 cancers-15-02538-t002:** Group assignments and dose levels (rat study).

Type of irAE	Highest Grade		Time of Onset after Initiation ICI	Management		Time to First Significant Response of irAEs (If Responsive)	Outcome	
Gastrointestinal irAEs								
irHepatitis (n = 44)	1	n = 3	4.4 months (range: 0.25–48 months)	Continue ICI-therapy (no intervention)	n = 3	3.6 days (range: 1–7 days)	Resolved	n = 33
	2	n = 14						
	3	n = 17		Systemic cort icosteroids	n = 41		Improved, but not resolved	n = 4
	4	n = 10					Second-line treatment required	n = 7
irColitis (n = 50)	1	n = 0	2.8 months (range: 0.07–26 months)	Continue ICI-therapy (no intervention)	n = 2	4.8 days (range: 1–56 days)	Resolved	n = 35
	2	n = 17		Systemic steroids	n = 43		Improved, but not resolved	n = 3
	3	n = 28		Mesalazin	n = 1			
	4	n = 3		Unknown	n = 4		Second-line treatment required	n = 12
	5	n = 0						
	Unknown	n = 2						
irPancreatitis (n = 16)	2	n = 6	6.6 months (range: 0.5–25 months)	Continue ICI-therapy (No intervention)	n = 7	8.6 days (range: 1–28 days)	Resolved	n = 10
	3	n = 8						
				Interruption ICI-therapy	n = 2		Improved, but not resolved	n = 4
	4	n = 2						
				Systemic corticosteroids	n = 7		Ongoing	n = 2
	5	n = 0						
irGastritis (n = 3)	1	n = 0	4.3 months (range: 3–6 months)	Continue ICI-therapy (no intervention)	n = 1	3.0 days (range: 3.0–3.0 days)	Resolved	n = 2
	2	n = 2						
	3	n = 1						
				Systemic steroids	n = 2		Improved, but not resolved	n = 1
	4	n = 0						
	5	n = 0						
irCholecystitis (n = 1)	2	n = 0	2.0 months (range: 2–2 months)	Systemic corticosteroids	n = 1	n/a (no improvement)	No improvement	n = 1
	3	n = 0						
	4	n = 1						
	5	n = 0						
Pulmonary irAEs								
irPneumonitis (n = 11)	1	n = 1	6.4 months (range: 1–24 months)	Continue ICI-therapy (no intervention)	n = 1	7.2 days (range: 3 days–28 days)	Resolved	n = 7
	2	n = 8						
	3	n = 2		Systemic steroids	n = 10		Improved, but not resolved	n = 4
	4	n = 0						
	5	n = 0						
irSarcoidosis (n = 1)	n/a	n = 1	6.0 months (range: 6–6 months)	Inhalative corticosteroids	n = 1	7.0 days (range: 7–7 days)	Improved, but not resolved	n = 1
Endocrine irAEs								
irThyroiditis (n = 18)	1	n = 4	2.8 months (range: 0.25–11 months)	Continue ICI-therapy (no intervention)	n = 1	2.4 days (range: 1–28 days)	Resolved	n = 2
	2	n = 14						
				Systemic steroids	n = 1			
	3	n = 0					Ongoing insufficiency requiring hormone substitution	n = 16
	4	n = 0		Hormone substitution	n = 16			
	5	n = 0						
irHypophysitis (n = 19)	1	n = 0	4.0 months (range 0.25–21 months)	Permanently discontinuation ICI	n = 1	1.9 days (range: 1–14 days)	Resolved	n = 3
	2	n = 12						
	3	n = 7		Hormone substitution	n = 17		Ongoing insufficiency requiring hormone substitution	n = 16
	4	n = 0		Unknown	n = 1			
	5	n = 0						
irAdrenalitis (n = 4)	1	n = 0	2.9 months (range 1.5–4 months)	Hormone substitution	n = 4	1.0 day (range: 1–1 day)	Ongoing insufficiency requiring hormone substitution	n = 4
	2	n = 1						
	3	n = 3						
	4	n = 0						
	5	n = 0						
irDiabetes (n = 1)	n/a	n = 1	3 months (range 3–3 months)	Hormone substitution	n = 1	1.0 day (range 1–1 day)	Ongoing insufficiency requiring hormone substitution	n = 1
Renal irAEs								
irNephritis (n = 6)	3	n = 5	2.7 months (range 0.25–4 months)	Systemic corticosteroids	n = 6	5.5 days (range 3–14 days)	Resolved	n = 5
	4	n = 1						
							improved, but not resolved	n = 1
	5	n = 0						
Cardiac irAEs								
irMyocarditis (n = 7)	2	n = 4	4.3 months (range 0.5–22 months)	Interruption of ICI therapy	n = 1	8.7 days (range 3–14 days)	Resolved	n = 6
	3	n = 3						
	4	n = 0		Systemic corticosteroids	n = 6		Second-line treatment required	n = 1
	5	n = 0						
Cutaneous irAEs								
irDermatitis (n = 22) (maculopapular rash n = 15, lichenoid rash n = 4, bullous pemphigoid n = 2, alopecia n = 1)	1	n = 16	4.2 months (range 0.1–21 months)	Continue ICI-therapy (no intervention)	n = 1	9.5 days (range 1–28 days)	Resolved	n = 12
	2	n = 4						
							Improved, but not resolved	n = 5
				Topical corticosteroids	n = 17			
	3	n = 1					Not improved	n = 2
	n/a	n = 1		Systemic corticosteroids	n = 4			
							Second-line treatment required	n = 3
Rheumatoid irAEs								
irArthritis (n = 10)	1	n = 0	3.3 months (range 0.03–11 months)	NSAIDs	n = 1	5.5 days (range 1–21 days)	Resolved	n = 3
	2	n = 10		Systemic corticosteroids	n = 7		Improved, but not resolved	n = 5
	3	n = 0		Systemic + intralesional corticosteroids	n = 2		Second-line treatment required	n = 2
Ophthalmic irAEs								
Ophthalmic irAEs (n = 3) (uveitis n = 1, conjunctivitis n = 1, papillitis n = 1)	1	n = 0	9.6 months (range 9–11 months)	Intravitreal corticosteroids	n = 1	8.0 days (range 3–14 days)	Resolved	n = 3
	2	n = 2						
				Topical corticosteroids	n = 1			
	3	n = 0						
	4	n = 0		Interruption of ICI-therapy (no intervention)	n = 1			
	n/a	n = 1						
Neurologic irAEs								
Neurologic irAEs (n = 11) (myositis n = 1, myasthenia gravis n = 1, myalgia n = 1, CPK increased n = 1, polyneuritis n = 1, vocal cord paresis n = 1, bilateral vestibulopathy n = 2, limbic encephalitis n = 1, peripheral sensory polyneuropathy n = 2)	1	n = 1	2.1 months (range 1.5–5 months)	Continue ICI-therapy (no intervention)	n = 2	7.0 days (range1–14 days)	Resolved	n = 9
	2	n = 4		Discontinuation ICI therapy (no intervention)	n = 2			
				Systemic corticosteroids	n = 4			
	3	n = 3		Systemic corticosteroids + intralesional corticosteroids	n = 1		Improved, but not resolved	n = 2
	4	n = 0						
				systemic corticosteroids (high dose)	n = 2			
	5	n = 0						
	n/a	n = 3						
Other irAEs								
CD4 + -count-decreased n = 1, polyserositis n = 1	1	n = 0	4.5 months (range 3–6 months)	permanent interruption of ICI therapy	n = 1	7 days (range 7–7 days)	resolved	n = 1
	2	n = 0						
	3	n = 1						
	4	n = 1		systemic corticosteroids	n = 1		no improvement	n = 1

**Table 3 cancers-15-02538-t003:** Incidence of sr/sd-irAEs.

Type of irAE	Total Number of irAEs	irAEs Treated with Systemic Steroids	sr/sd-irAEs	Percentage of sr/sd-irAEs (%)
irHepatitis	44	41	11	26.8
irColitis	50	43	15	34.9
irPancreatitis	16	7	1	14.3
irGastritis	3	2	0	0
irPneumonitis	11	10	4	40.0
irNephritis	6	6	1	16.7
irMyocarditis	7	6	1	16.7
irDermatitis	22	4	3	75.0
irArthritis	10	9	7	77.8
Neurologic irAEs	11	7	2	25.6

**Table 4 cancers-15-02538-t004:** Second-line treatment.

irAE	Second-Line Treatment	Mean Time to First Response of irAE	Outcome
irColitis (n = 12)	Infliximab 5 mg/kg 1 dose (n = 6) 2 doses (n = 6)	1.3 days (0 days–7 days)	Complete resolution (n = 9) Improvement, but not resolution (n = 1) Third-line therapy (n = 2)
irHepatitis (n = 7)	Mycophenolate-mofetil 1g/day (n = 2) 2g/day (n = 5)	2.8 days (2 days–7 days)	Complete resolution (n = 5) Improvement, but no resolution (n = 1) Third-line therapy (n = 1)
irDermatitis (lichenoid rash n = 2, bullous drug reaction n = 1)	Acitretin 10 mg xtracorporeal photopheresis IVIG 2g/kg	4 weeks 6 weeks 3 weeks	Improvement, but no resolution (n = 3)
irArthritis (n = 2)	Methotrexate 15 mg/week s.c.	6 weeks–3 months	Improvement, but no resolution (n = 2)
irMyocarditis (n = 1)	IVIG 2g/kg	2 weeks	Resolved

**Table 5 cancers-15-02538-t005:** Recommendation for second-line therapy compared with cohort. (Abbreviations: ECP = extracorporeal photopheresis; TNF = tumor necrosis factor; IL = interleukin; IVIG = intravenous immunoglobulins.)

	Type of irAE	ASCO Guidelines	NCCN Guidelines	ESMO Guidelines	Patient Cohort
Skin Toxicities	Maculopapular rash			Infliximab, tocilizumab	
	Bullous dermatoses	Rituximab			IVIG (1/1)
	Other cutaneous adverse reactions	IVIG/cyclosporine			
	irLichen ruber				Acitretin (1/2), ECP (1/2)
Gastrointestinal Toxicities	irColitis	Infliximab/TNF-α blocker, Vedolizumab if refractory to TNF-α blocker	Infliximab, vedolizumab	Infliximab, if not responsive, higher-dose infliximab, ustekinumab, tofacitinib, extracorporeal photopheresis, fecal microbiota transplantation	Infliximab (12/12)
	irHepatitis	Grade 3: mycophenolate mofetil/azathioprine Grade 4: mycophenolate mofetil (avoid infliximab)	Mycophenolate mofetil (infliximab not recommended)	Mycophenolate mofetil, tocilizumab, tacrolimus, azathioprine, cyclosporine, antithymocyte globulin	Mycophenolate mofetil (7/7)
	irPancreatitis				
Lung Toxicity	irPneumonitis	Infliximab/mycophenolate mofetil/IVIG/cyclophosphamide	Infliximab, mycophenolate mofetil, IVIG	Tocilizumab, infliximab, IVIG, mycophenolate mofetil, cyclophosphamide	
Musculoskeletal Toxicities	irArthritis	Methotrexate, leflunomide, TNF-α inhibitor, IL-6 receptor inhibitor (not to be used in patients with colitis)	Infliximab, methotrexate, tocilizumab, sulfasalazine, azathioprine, leflunomide, IVIG	Methotrexate, azathioprine, mycophenolate mofetil, tacrolimus,	Methotrexate (2/2)
	irMyositis	Plasmapheresis, IVIG, methotrexate, azathioprine, mycophenolate mofetil		Tocilizumab, TNF-α inhibitor	
	Polymyalgia-like syndrome	Methotrexate, IL-6 receptor inhibition (not to be used in patients with colitis)			
Renal Toxicities	irNephritis	Mycophenolate mofetil	Azathioprine, cyclophosphamide, infliximab, mycophenolate mofetil		
Nervous System Toxicities	irMyasthenia gravis	IVIG/plasmapheresis		Abatacept, tacrolimus, cyclophosphamide, rituximab, infliximab, tocilizumab, azathioprine, mycophenolate mofetil	
	irGuillain–Barré syndrome	IVIG/plasmapheresis	IVIG, plasmapheresis	Plasmapheresis, IVIG	
	Peripheral neuropathy	According to Guillain-Barré syndrome management	According to Guillain–Barré syndrome management	IVIG	
	Aseptic meningitis				
	Encephalitis	IVIG/rituximab/plasmapheresis	IVIG, rituximab		
	Transverse myelitis	IVIG	IVIG, plasmapheresis	Plasmapheresis	
Hematologic Toxicities	irHemolytic anemia	Rituximab, IVIG, cyclosporine A, mycophenolate mofetil			
	irThromocytopenic purpura	Plasmapheresis, rituximab			
	irAplastic anemia	Horse antithymocyte globulin + cyclosporine A, if no response, rabbit antithymocyte globulin + cyclosporine/cyclophosphamide			
	irThromocytopenia	IVIG, rituximab		IVIG, eltrombopag	
	irHemophilia	Rituximab/cyclophosphamide, cyclosporine, immunoadsorption			
	irAgranuloytosis				
	irHematophagocytic lymphohistiocytosis			Tocilizumab	
Cardiovascular Toxicities	irMyocarditis, irPericarditis	Mycophenolate mofetil, infliximab, antithymocyte globulin	Antithymocyte globulin, infliximab, IVIG, mycophenolate mofetil	Tocilizumab, mycophenolate mofetil, antithymocyte globulin, alemtuzumab, abatacept	IVIG (1/1)

## Data Availability

All data generated or analyzed during this study are included in this article. Further enquiries can be directed to the corresponding author upon reasonable request.

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
