# Peer review of "Steroid-Refractory Immune-Related Adverse Events Induced by Checkpoint Inhibitors"

_cancers, 2023, doi:10.3390/cancers15092538_

Round 1

Reviewer 1 Report

The review entitled 'Second-line immunosuppressants in steroid-refractory immune-related adverse events in patients treated with immune checkpoint inhibitors’ is a very important study and outlines the adverse events related to use the Immunotherapy and with a particular focus on check point inhibitor therapy. The study is very detailed. The introductory section related to the overview of checkpoint inhibitors is very interesting.

The authors could improve the review article.

1.      The table shown in Material and methods section is missing some key information, such as the number of patients that have taken part in the study. The authors need to describe that 181 patients were only included in the study as the documentation was available. The remaining 225 patients can be described as not considered due to lack of documentation. The table should include a table legend describing this information and table number.

2.      The authors can describe about the data source that was collected as part of the study in the materials section.

3.      Whether the study included population for a specific region of the country or there is a diverse group included in the study.

4.      What is the follow up of the patients involved in the study, how many visits per month?

5.      What is the survival rate, a Kaplan Meier plot would be helpful.

6.      The authors can include a decision cascade of administration of various therapies such as drugs, monoclonal antibodies, etc due to adverse events.

7.      What are the potential symptoms of the patients upon onset of these adverse events?

8.      Other therapeutic interventions that could be potentially explored for treatment can be outlined.

9.      The authors need to include the latest developments in the field and discuss more broadly the upcoming technologies that could pave way for future medicinal approaches for treatment of these patients.

           10. The authors can describe on the emerging targets in immunotherapy such as mRNA therapy or CRISPR mediated Immunotherapy. 

Author Response

please see the attachement

Reviewer 2 Report

Following are the suggestion for the authors that they may include if necessary;

1) In title, there are 2 time the word immune, so better to modify the title.

2) Line 21, there is no any meaning of sentence, using two time was, modify this statement.

3) In method section, I suggest you to read the article related to the topic and modify the method. Method section is incomplete. Sample size, design and how to calculate the sample is missing. Further, where the study was taken. Consent section is also missing. Overall, the method section is weak. It be elaborated. Further, how adverse drug reactions are detected. Which scale was used.???

4) Limitation of the study is also missing.

5) Tables are too large which is difficult to understand. So try to break the tables in different sections and elaborate individual. Further, statistical part is also missing the study.

Round 2

Reviewer 2 Report

I have read the article. After major modification, it seems ok now in terms of introductio, method, discussion and results. The method section is now improved after changes.